# Regulation of Cecal Microbiota and Improvement of Blood Lipids Using Walnut Non-Dairy Creamer in High-Fat Mice: Replacing Traditional Non-Dairy Creamer

**DOI:** 10.3390/molecules29184469

**Published:** 2024-09-20

**Authors:** Mingming Wang, Feng Zhang, Chunlei Tan, Si Huang, Hongyu Mu, Kuan Wu, Yinyan Chen, Jun Sheng, Yang Tian, Ya Wang, Cunchao Zhao

**Affiliations:** 1College of Food Science and Technology, Yunnan Agricultural University, Kunming 650201, China; wmm295813@163.com (M.W.); 15762196909@163.com (F.Z.); tcl98316@163.com (C.T.); 18870918872@163.com (S.H.); muhongyu0710@163.com (H.M.); wink577@163.com (K.W.); 18214081576@163.com (Y.C.); shengjunpuer@163.com (J.S.); tianyang1208@163.com (Y.T.); 2State Key Laboratory of Tea Plant Biology and Utilization, School of Tea and Food Science & Technology, Anhui Agricultural University, Hefei 230036, China; 3Yunnan Plateau Characteristic Agricultural Industry Research Institute, Kunming 650201, China; 4Engineering Research Center of Development and Utilization of Food and Drug Homologous Resources, Ministry of Education, Yunnan Agricultural University, Kunming 650201, China; 5Pu’er University, Pu’er 665000, China; 6College of Science, Yunnan Agricultural University, Kunming 650201, China; 7Key Laboratory of Precision Nutrition and Personalized Food Manufacturing, Ministry of Education, Yunnan Agricultural University, Kunming 650201, China

**Keywords:** non-dairy creamer, high-fat mice, lowering blood lipids, intestinal flora

## Abstract

Non-dairy creamer is a class of microencapsulated powdered fats and oils that are widely used in the food industry. However, the oils used in it are hydrogenated vegetable oils, which contain large amounts of saturated fatty acids and are extremely harmful to the human body. This study investigated the effects of replacing hydrogenated vegetable oil with walnut oil to prepare walnut non-dairy creamer on lipid levels and intestinal microorganisms in mice. The results show that low-dose walnut non-dairy creamer significantly decreased the contents of TC and TG in serum and increased the content of HDL-C (*p* < 0.01). The contents of MDA, ALT, and AST were significantly decreased, while the content of SOD was increased (*p* < 0.01). The abundance of *Firmicutes* in the walnut non-dairy creamer group decreased, and the abundance of *Bacteroidetes/Firmicutes* (B/F) increased, which significantly increased the richness of *Lactobacillus* and *Oscillospira* (*p* < 0.01). *Allobaculum* richness was significantly decreased (*p* < 0.01). In conclusion, a low dose of walnut non-dairy creamer can effectively promote the metabolism of blood lipids in vivo, alleviate oxidative stress injury and lipid accumulation damage to mouse hepatocytes, and ameliorate the adverse effects of a high-fat diet on the intestinal microbiota of mice. This study provides a theoretical basis for the replacement of traditional non-dairy creamer and the research and development of walnut deep processing.

## 1. Introduction

Non-dairy creamer, also known as coffee creamer, is a powder containing syrup, hydrogenated vegetable oil, sodium caseinate, etc., which is formed by emulsification, homogenization, and spray drying [1,2]. Due to its good stability, convenient storage and transportation, and low price, it is widely used in coffee and other beverages [3], giving them an ideal flavor and texture [3,4]. However, the oil used in the current market is hydrogenated vegetable oil, which contains a large amount of saturated fatty acids, and long-term consumption will lead to various lipid disorders and increase the risk of cardiovascular diseases and obesity [5,6,7]. Hanna et al. [8] found that feeding non-dairy creamers significantly increased triglycerides (TGs), low-density lipoprotein (LDL-C), and malondialdehyde (MAD), as well as significantly decreased Glutathione peroxidase (GSH-PX) and insulin levels and increased the risk of cardiovascular disease in rats. Therefore, it is of great importance to find a new nutritious oil to replace hydrogenated vegetable oils for the preparation of healthy non-dairy creamers.

Walnut oil (WO), as a high-quality vegetable oil, contains a large amount of monounsaturated fatty acids (MUFAs), such as oleic acid, and polyunsaturated fatty acids (PUFA)s, such as α-linolenic acid and linoleic acid, which have antioxidant and anti-inflammatory effects and can prevent atherosclerosis, hypertension, and other chronic diseases [9,10]. Among them, linolenic acid (C18:3 n-3) is generally considered to be particularly beneficial to the human body; it can reduce low-density cholesterol to a certain extent and prevent arteriosclerosis [11]. Katsri et al. [12] found that the use of oleic acid-rich vegetable oils instead of saturated fats reduced serum cholesterol and low-density lipoprotein (LDL) blood levels. Changes in the gut microbiota are another factor in the development of obesity disorder. Many studies have shown that the intestinal microbiota plays an important role in the occurrence and maintenance of obesity [13], that obesity can be interfered with by regulating the intestinal microbiota [14], and that dietary oil can be metabolized and utilized by the intestinal microbiota to produce metabolites beneficial to body health. Ye, et al. [15] found that moderate amounts of ω-6/ω-3 PUFA can maintain normal intestinal permeability, improve its diversity and structure, and enable the host to produce beneficial metabolites. Fujun Miao [16] and others found that walnut oil exerted anti-inflammatory effects on murine colitis by inhibiting NLRP3 inflammasome activation and modulating the intestinal microbiota. In addition, there is evidence that n-3-PUFA-rich oils can inhibit NLRP3 inflammatory vesicle activity and modulate the intestinal flora [17]. Currently, little is known about the use of walnut oil in non-dairy creamers. However, non-dairy creamers with walnut oil as a core material have great potential not only for preventing the accumulation of body fat, but also for avoiding health problems caused by excessive intake of saturated fatty acids. Therefore, in this study, non-dairy creamer was prepared from walnut oil, focusing on the effects of walnut non-dairy creamer on blood lipid levels and intestinal flora in hyperlipidemic mice, which provided a theoretical basis for the substitution of traditional non-dairy creamer.

## 2. Results and Discussion

### 2.1. Effect of Different Non-Dairy Creamers on Body Weight and Food Intake in High-Fat Mice

As can be seen from Table 1 and Figure 1C,D, there was no significant difference in body weight among all groups at the beginning of the experiment (*p* > 0.05). After 12 weeks of feeding a high-fat diet, the body weight of mice in the HDF group was significantly higher than that in the CK group (*p* < 0.05). Compared with the HDF group, the body weight of mice in the L-MCC and H-MCC groups was significantly decreased (*p* < 0.05), and the body weight of mice in the H-NDC group was significantly increased (*p* < 0.05). Compared with the HDF group, there was no significant difference in food intake in the L-MCC and H-MCC groups (*p* > 0.05). In summary, feeding MCC can significantly reduce weight gain induced by a high-fat diet.

### 2.2. Effect of Different Non-Dairy Creamers on Glucose Tolerance in High-Fat Mice

As can be seen from Figure 1A, the OGTT curves of the mice in each group showed an increasing and then decreasing trend, and the peak in their blood glucose concentration was at 30 min after the administration of the drug. The blood glucose level in the HFD group was higher than that in the CK group at all stages. As can be seen from Figure 1B, the AUC value of the HFD group was also significantly higher than that of the CK group (*p* < 0.01) and was not significantly different from that of the L-NDC and H-NDC groups (*p* > 0.05). Compared with the HDF group, the area-on-the-curve (AUC) level of the L-MCC group was significantly lower (*p* < 0.01), and the blood glucose levels of the groups stabilized after 120 min of glucose gavage. It has been shown that a high-fat diet disrupts glucose metabolic homeostasis in mice, resulting in impaired glucose clearance [18,19]. The OGTT results indicated that prolonged gavage of MCC at a certain range of concentrations significantly enhances glucose tolerance in mice induced by a high-fat diet.

### 2.3. Effect of Different Non-Dairy Creamers on TC, TG, HDL-C, and LDL-C Content in High-Fat Mice

Walnut oil is rich in linoleic acid, linolenic acid, and other unsaturated fatty acids, which, when absorbed, can effectively reduce the human blood cholesterol content, prevent atherosclerosis, and improve the function of the blood vessel wall [20,21]. As can be seen from Figure 2A,B, the serum TG content of HDF mice was 1.86 mmol/L-1 and the TC content was 4.74 mmol/L, which were significantly higher than those of the CK group (*p* < 0.05), suggesting that the modeling of the high-fat model was successful [22]. L-MCC significantly reduced the levels of blood TG and TC in mice (*p* < 0.05), the levels of TC and TG were elevated in mice in the NDC group, and the levels of TG were significantly higher compared with those in the HDF group (*p* < 0.05). As can be seen from Figure 2C,D, compared with the HDF group, MCC reduced the LDL-C level, but the difference was not significant (*p* > 0.05), and the NDC group also showed the same effect. Compared with the HDF group, L-MCC could significantly increase serum HDL-C levels in mice on the high-fat diet (*p* < 0.05), but there was no significant difference in high doses (*p* > 0.05). This indicates that gavage of MCC in a certain concentration range can regulate the LDL-C and HDL-C content of mice, and with the increase in the concentration of gavage, it is difficult to eliminate the energy from walnut oil’s action. And the NDC group could significantly increase the content of HDL-C (*p* < 0.05). The above results indicate that L-MCC could effectively regulate the blood lipid level of mice on a high-fat diet.

### 2.4. Effect of Different Non-Dairy Creamers on Antioxidant Indices in High-Fat Mice

Total superoxide dismutase (SOD) can effectively remove oxygen free radicals in the body and maintain the balance between oxidation and antioxidant capacity of the body, which is one of the main indexes of the body’s antioxidant capacity [23]. As can be seen from Figure 3A, compared with the CK group, the HDF group extremely significantly reduced the SOD activity in high-fat mice (*p* < 0.01). After the application of MCC and NDC, the SOD level of the mice was significantly increased compared with that of the HDF group (*p* < 0.01), and the effect in the MCC groups was better. There was no significant difference in SOD levels between MCC groups (*p* > 0.05). SOD levels decreased significantly with increasing gavage doses between NDC groups (*p* < 0.01).

MDA is a marker of lipid peroxidation damage to any cell membrane, which can reflect the rate and intensity of lipid peroxidation in the organism, and also indirectly reflects the degree of peroxidative damage to tissues [24]. As can be seen in Figure 3B, the high-fat diet significantly increased the MDA level of mice in the HDF group compared with those in the CK group (*p* < 0.01). Compared with the HDF group, both MCC groups significantly reduced MDA (*p* < 0.01), and there was no significant difference between the two groups (*p* > 0.05). Zhang [25] found that hazelnut oil microcapsules can also reduce the level of lipid peroxidation products, which is consistent with our research results. MDA content in the NDC group was significantly increased (*p* < 0.01). In summary, MCC can effectively reduce the damage of lipid oxidative stress to cells and tissues.

GSH-PX is an enzyme that catalyzes the decomposition of H_2_O_2_, scavenging lipid peroxides and blocking the chain reaction of lipid peroxidation, which leads to the alleviation of hepatocellular damage [24]. As can be seen in Figure 3C, the HDF group fed a high-fat diet had a significantly reduced GSH-PX content compared to the CK group (*p* < 0.01). After MCC intervention, the content of GSH-PX in mice was increased, but the difference was not significant (*p* > 0.05). With the HDF group, the GSH-PX content was reduced in both NDC groups. In summary, the effect of NDC was not as good as that of MCC.

### 2.5. Effect of Different Non-Dairy Creamers on ALT and AST Levels in High-Fat Mice

ALT and AST are predominantly found in hepatocytes and their activity can often be used to determine liver injury or liver dysfunction [26]. ALT and AST are dysfunctional enzymes in serum and plasma. Enzyme activity increases when ALT and AST are released into the bloodstream when hepatocytes are damaged, and this increase in ALT and AST activity is a sign of liver damage [27]. As can be seen in Figure 3D,E, ALT and AST levels were significantly higher (*p* < 0.01) in the HDF group compared to the CK group, indicating that long-term feeding of high-fat chow leads to liver damage in mice. The serum levels of ALT and AST in high-fat mice were significantly reduced after MCC intervention (*p* < 0.01). The L-NDC group also significantly reduced AST levels (*p* < 0.01), but the effect was not as good as that of MCC. In addition, the H-NDC group had no significant effect on ALT and AST levels (*p* > 0.05). The above results indicate that MCC can effectively reduce liver injury in high-fat mice.

### 2.6. Effect of Different Non-Dairy Creamesr on Liver Cells in High-Fat Mice

The liver plays an important role in lipid metabolism and its injury can lead to lipid accumulation and a range of metabolic disorders [28]. As can be seen from Figure 4, the hepatocytes of mice in the CK group were relatively neatly arranged and tightly packed, and there was no significant dilatation or extrusion of the hepatic sinusoids. In the HDF group, the hepatocytes were loosely arranged and disorganized, and the cytoplasm of hepatocytes contained lipid vacuoles of varying sizes (black arrow) as a hallmark of steatosis. This was attributed to the fact that high-fat dietary feeding led to the accumulation of lipids in the hepatocytes of the mice, which in turn led to the development of steatosis. Compared with the HDF group, the liver symptoms of L-MCC and H-MCC groups were reduced. In the L-MCC group, a small number of hepatocytes showed slight granular degeneration, loose cytoplasm, and low staining intensity (yellow arrow), while in the H-MCC group, a small number of hepatocytes showed steatosis (black arrow). A large amount of hepatocyte mild steatosis (black arrow) was seen in the L-NDC and H-NDC groups, and small circular vacuoles were seen in the cytoplasm. In addition, in the H-NDC group, a small number of necrotic hepatocytes with punctate necrosis (red arrowheads) were seen, with deeply stained and fragmented nuclei, surrounded by a large amount of granulocyte infiltration (blue arrowheads). These results indicate that MCC intervention can effectively alleviate liver lipid accumulation caused by a high-fat diet and can effectively improve liver damage.

### 2.7. Effect of Different Non-Dairy Creamers on the Structure of Intestinal Flora in High-Fat Mice

#### 2.7.1. Venn Diagram Analysis Based on ASV/OUT Data

The gut is an important place for the body to absorb nutrients and the largest immune organ, and the mutually beneficial symbiotic relationship between intestinal microbes and their hosts is mediated by the gut [29]. As an important part of the intestinal environment, intestinal flora play a crucial role in regulating intestinal barrier function [30]. To study the effect of different non-dairy creamer on the intestinal flora of mice, the cecum flora of high-fat mice were analyzed by 16S rRNA sequencing. As can be seen in Figure 5A, a total of 3504 OTUs (Operational Classification Units) were generated for all samples, of which 91 OTUs were common to the control and experimental groups, and 937 (26.74%), 226 (6.45%), 377 (10.76%), 577 (15.75%), 577 (16.47%), and 289 (8.25%) were unique to the CK, HDF, L-MCC, H-MCC, L-NDC, and H-NDC groups, respectively. Differences between the number of ASVs in each group indicate differences in the similarity of the intestinal flora of the mice in each group.

#### 2.7.2. Alpha Diversity Analysis

In order to assess the alpha diversity of the microbial community in a more comprehensive way, Chao1, Shannon, Simpson, and Pielou indices were measured and calculated, where Chao1 and Observed species indices characterize richness, Shannon and Simpson indices characterize diversity, and Pielou ‘s evenness index characterizes evenness [31]. As can be seen from Figure 5B–F, Chao1, Shannon, Simpson, Pielou, and Simpson index richness were significantly reduced in the HDF compared to the CK group (*p* < 0.05). Compared with the HDF group, the MCC group all significantly increased the Chao1, Shannon, Simpson, and Pielou index richness (*p* < 0.05), and the effect was better at lower doses. L-NDC significantly increased the diversity index (*p* < 0.05), but showed the opposite trend with the increase in dose.

#### 2.7.3. Beta Diversity Analysis

This study evaluates Beta diversity among different samples based on Bray–Curtis distance (BD) using principal coordinate analysis (PCA) and non-metric multidimensional scaling analysis (NMDS). As can be seen from 5G, the samples from each group were essentially in one area, indicating good reproducibility of samples within groups and separation of groups from each other. These results indicate that feeding a high-fat diet and non-dairy creamer significantly altered the composition of gut microbes in mice compared to feeding a basal diet. As can be seen from Figure 5H, the MCC group can be separated from the HDF group, and the distance is relatively far. The distance between the NDC group and the HDF group was relatively close, and there was a partial overlap between the H-NDC group and the HDF group, indicating that the intestinal microbial composition of the two groups was relatively close. The distance between group MCC and group CK is relatively close. Compared with the H-MCC group, the L-MCC group is closer to the CK group. In conclusion, MCC intervention can effectively improve the intestinal flora composition of high-fat mice.

#### 2.7.4. Phylum Level Analysis

Changes in the intestinal flora are one of the reasons for the formation of obesity in the organism [32]. Figure 6A shows the histogram of species distribution of mouse intestinal microorganisms at the phylum level in different groups. As can be seen from Figure 6, *Bacteroidetes*, *Firmicutes*, *Proteobacteria*, *Actinobacteria, Verrucomicrobia*, etc., are the major groups distributed in the mouse intestines, accounting for 90% of the intestinal microbiota. *Verrucomicrobia* is the main flora distribution in the mouse intestine, accounting for more than 90% of the intestinal microbiota. As can be seen from Figure 6A,C–F, *Bacteroidetes*, *Firmicutes*, *Proteobacteria*, *Actinobacteria*, and *Verrucomicrobia* are the main microflora in the intestinal tract of mice, accounting for more than 90% of the intestinal microflora. The pathogenesis of obesity is strongly associated with an increased ratio of B/F [33]. Our results showed that the HFD group significantly increased *Firmicutes* and reduced *B/F* content (*p* < 0.01) compared to the CK group. All other test groups significantly reduced the thick-walled *Firmicutes* (*p* < 0.01) and alleviated the reduction in *B/F* content. This is the same as the results of Zhang et al. [34]. It has been shown that *Verrucomicrobia* can produce short-chain fatty acids, such as propionic acid and butyric acid, which are important for intestinal health and immune system regulation [35]. In our study, we found that L-MCC significantly increased the abundance of *Verrucomicrobia* (*p* < 0.01); there was no significant difference in the H-MCC group (*p* > 0.05), and H-NDC significantly decreased the abundance of *Verrucomicrobia* (*p* < 0.01). This indicated that consumption of a certain dose of MCC could effectively improve the problem of intestinal flora dysbiosis caused by a high-fat diet.

#### 2.7.5. Genus Level Analysis

As can be seen from Figure 6G–I, the relative abundance of *Lactobacillus* and *Oscillospira* in the HDF group was significantly lower (*p* < 0.01) and the abundance of *Akkermansia* did not change significantly (*p* > 0.05) compared to the CK group. Studies have shown that Lactobacillus can maintain the balance of flora, improve digestion, and enhance immunity [36]. In addition, *Oscillospira*, as a candidate for next-generation probiotics, has the effects of weight loss, lowering lipid levels, and alleviating metabolic syndrome, and its unique weight-loss and lipid-lowering activity may be related to its ability to produce short-chain fatty acids such as butyrate [37]. In our study, we found that the L-MCC group significantly increased the abundance of *Oscillospira*, *Lactobacillus*, and *Akkermansia* (*p* < 0.01). However, the almost opposite trend was shown with increasing doses. The L-NDC group significantly increased the abundance of *Lactobacillus* and *Oscillospira* (*p* < 0.01), but the effect was not as good as that of the MCC group. It suggests that gavage of MCC in a certain concentration range helps to increase the beneficial microorganisms in the intestinal tract of mice on a high-fat diet and improve the adverse effects of high-fat diets on the intestinal microorganisms of mice.

#### 2.7.6. Analysis of Interspecific Differences in Microbial Communities

LEfSe is used to detect features with significant abundance differences and to find taxa that are significantly different from the abundance [38]. As can be seen from Figure 6J, the evolutionary branching diagram demonstrates microorganisms with intergroup differences in each taxonomic level, representing the five taxonomic levels of the genus, family, order, and phylum from the outside of the circle inwards, with the yellow dots representing microorganisms with no significant differences, and the other colors representing microorganisms that are important for the corresponding group. The effects of specific bacterial taxa were compared using linear discriminant analysis (LDA). As can be seen from Figure 6K, although the results showed no difference in most bacteria, some specific bacteria were found and most of the specific taxa were from the L-MCC group, which indicated that the low dose of non-dairy creamer had a strong effect on the gut microbiota. The dominant bacteria in the CK group were *Actinobacteria*, *Bifidobacteria*, and *Lactobacillus* (LDA > 3). Compared with the CK group, the dominant bacteria in the HDF group was *Ruminococcus* (LDA > 4). Studies have shown that Ruminococcus is more abundant in patients with obesity than in people of normal weight [39]. The dominant bacteria in the L-MCC group were *Verrucomicrobiaceae*, *Mucinophilic Akkermansia*, and *Bacteroidetes*. The colonization of *Akkermansi a* in the intestine is closely related to the health of the host, which can improve adverse symptoms such as obesity, insulin resistance, and glucose tolerance; regulate the immune response of the body; and maintain the metabolic balance in the body [40]. Compared with the CK group, the dominant flora in L-NDC and H-NDC were *Clostridium* and *Desulfovibrio*, respectively. *Desulfovibrio*, as a harmful bacterium in humans, not only increases the production of intestinal secondary bile acids, but also increases the hydrophobicity of bile acids which, in turn, promotes the absorption of cholesterol from the intestinal tract, leading to an increase in cholesterol load in tissues such as the liver and promotes the secretion of cholesterol from hepatocytes into the bile [41]. In summary, although there were large differences in specific genera between the groups, the MCC intervention moderated the high-fat-diet-induced intestinal flora disruption to varying degrees, increasing the abundance of beneficial intestinal bacteria, such as *Verrucomicrobiaceae*, *Akkermansia*, and *Bacteroidetes*, whereas the NDC led to an increase in the abundance of harmful bacteria, such as *Desulfovibrio* harmful bacteria abundance.

#### 2.7.7. Correlation Analysis of Microorganisms with Lipid Metabolism Indicators

In order to further reveal the correlation between gut microbes and obesity-related parameters induced by a high-fat diet, key genera of gut microbes were correlated with obesity-related parameters in this study. As can be seen from Figure 7, *Desulfovibrio*, *Ruminococcus*, and *Proteobacteria* all showed a positive correlation with body weight, and controlling the increase in these bacteria may help to control weight gain. In the previous results, we showed that MCC intervention significantly reduced *Desulfovibrio* and *Ruminococcus* compared to NDC (*p* < 0.01), and *Lactobacillus* showed a negative correlation with TC, TG, MDA, LDL-C, AST, ALT, SOD, HDL-C, and body weight except for GSH-PX and was significantly negatively correlated with LDL-C and MDA. Our study showed that MCC significantly increased the abundance of Lactobacillus (*p* < 0.01).

## 3. Materials and Methods

### 3.1. Materials and Reagents

Commercial non-dairy creamer (the main components are hydrogenated palm oil, glucose syrup, sodium caseinate, maltodextrin, monoglyceride, etc.) is a powder oil prepared by emulsification, mixing, homogenization, spray drying, and other processes. Its products were purchased from China Tianjin Zhengchi International Trading Co., Ltd., Tianjin, China. Walnut non-dairy creamer (the main components are walnut oil, glucose syrup, sodium caseinate, maltodextrin, monoglyceride, etc.) is a powdered oil prepared by emulsification, mixing, homogenization, spray drying, and other processes. Products were purchased from Yunnan Kunming Institute of Biological Products Co., Ltd., Kunming, China.

### 3.2. Fatty Acid Composition

Fatty acid composition refers to the previous research of the laboratory [42]; the specific fatty acid composition is as follows: the saturated fatty acid content of commercial non-dairy creamer is 99.72%, the monounsaturated fatty acid content is 0.16%, and the polyunsaturated fatty acid content is 0.12%. The unsaturated fatty acid content of walnut non-dairy creamer is as high as 87.11%, the monounsaturated fatty acid content is 25.95%, and the polyunsaturated fatty acid content is 61.16%.

### 3.3. Animals and Experimental Design

Sixty specific pathogen-free (SPF) C57BL/6J male mice were selected, and experimental procedures involving animals were carried out according to the recommendations of Yunnan Agricultural University’s Guide for the Care and Use of Laboratory Animals (Approval No. 202209012). All mice were housed in the SPF Animal Facility under controlled conditions (temperature 22–24 °C, relative humidity 50–60%, and a light/dark cycle of 12 h). After one week of acclimatization feeding, the mice were randomly divided into 6 groups of 10 mice each: blank control group (CK), high-fat model group (HFD), low-dose group of walnut non-dairy creamer (L-MCC), high-dose group of walnut non-dairy creamer (H-MCC), commercial non-dairy creamer low-dose group (L-NDC), and commercial non-dairy creamer high-dose group (H-NDC). The gavage dose was set according to the study of Qi et al. [43]. The feeding and gavage protocols for mice in different treatment groups are shown in Table 2. The feed used for the mice in the CK group was the experimental basal diet (10% fat, 15% protein, and 75% carbohydrate), and the feed used for the HFD group with each non-dairy creamer-treated group was a high-fat diet (60% fat, 25% protein, and 15% carbohydrate). During the test period, the mice were gavaged once a day at a fixed time in the morning, fed and watered freely during the rest of the day, and the bedding was changed every 48 h. The drug was administered continuously for 12 weeks. Each mouse was weighed weekly and each mouse’s food intake was measured daily throughout the intervention trial. After 12 weeks, mice were euthanized by anesthesia after a 12 h fast, followed by cervical dislocation. Blood was collected from the eyeballs of each mouse, and after centrifugation of the blood samples for 15 min at 4 °C and 4000 rpm, serum was collected and stored at −80 °C until analyzed. The liver was collected, rinsed, weighed, and then frozen in liquid nitrogen and stored at −80 °C for further use. In addition, a portion of the liver tissue was fixed in 10% formalin and embedded in paraffin wax for histological study. The contents of the cecum were collected and immediately stored at −80 °C for further use.

### 3.4. Glucose Tolerance Test

An oral glucose tolerance (OGTT) test was used to evaluate the effect of walnut non-dairy creamer on glucose tolerance in mice fed a high-fat diet. The glucose tolerance test was conducted one week before the end of the experiment; mice were fasted for 12 h. Blood glucose measurements at the 0 min time point were performed first by tail-tip blood collection, then glucose (2 g/kg) was injected intraperitoneally, and measurements were taken at 30 min intervals after the end of the injection. The specific protocol was to measure tail vein blood glucose values using a glucometer at 30, 60, 90, and 120 min time points, and the area under the curve of glucose versus time (OGTT-AUC) was calculated using the following formula [44]:
(1)$$\begin{array}{c} \mathrm{OGTT}-\mathrm{AUC}=[({G0}_{h}+{G0.5}_{h})\times 0.5+({G0.5}_{h}+{G1}_{h})\times 0.5+({G1}_{h}\\ +{G1.5}_{h})\times 0.5+({G1.5}_{h}+{G2}_{h})\times 0.5]/2 \end{array}$$
where *G* is the blood glucose value.

### 3.5. Biochemical Assays

TG, C, HDL-C, LDL-C, SOD, GSH-PX, MDA, ALT, and AST were detected by using a test kit from Nanjing Institute of Biological Engineering, Nanjin, China.

### 3.6. Histopathological Observation of Liver Tissue

Fresh mouse liver tissue was placed in 10% formalin and handed over to Yunnan Huiteng Biotechnology Co., Ltd., Kunming, China. The tissue was dehydrated with 85% and 95% alcohol for 5 min and embedded in paraffin molds. Paraffin molds were sliced on tissue slides 4–5 µm thick and stained with the hematoxylin–eosin (HE) method: they were soaked in a hematoxylin solution for 3–5 min, washed in water, and then stained with the eosin solution for 5 min. Analysis of the tissue slides was performed using a standing light microscope (Nikon Eclipse E100, Nikon, Japan) and a Pannoramic 250 digital section scanner (3D HISTECH, Budapest, Hungary) for observation and photographs.

### 3.7. DNA Extraction and 16s rRNA Gene Sequencing

The cecum content samples were collected from cecum samples using a DNA extraction kit to amplify the V3–V4 region of the bacterial 16S rRNA gene based on the Illumina Miseq platform. The PCR primer sequences were 338F (5′-ACTCCTACGGGGAGGCAGCAG-3′) and 806R (5′-GGACTACHVGGGTWTCTAAT-3′). All PCR products were subjected to 2% agarose gel electrophoresis, and gene samples recovered and purified from the agarose gel using a DNA extraction kit were subjected to 2 × 250 bp bipartite sequencing on a MiSeq machine using an illumina novaseq6000 Reagent Kit V3 (600 cycles). Data were analyzed using QIIME2 dada 2 (2019.4) software.

### 3.8. Statistical Analysis

Statistical software SPSS 19.0 was used to conduct One-way ANOVA on the experimental data. The data were represented by Mean ± SD; *p* < 0.05 meant a significant difference, and *p* < 0.01 meant an extremely significant difference. SPSS was used for Pearson correlation analysis, and Origin 2019 software was used for mapping.

## 4. Conclusions

Our study shows that feeding MCC significantly alleviates weight gain induced by a high-fat diet. Low-dose MCC can effectively regulate the blood lipid level of hyperlipidemic mice and reduce the damage of cells and tissues caused by lipid oxidative stress. In addition, MCC significantly increased the abundance of beneficial bacteria such as *B/F*, *Lactobacillus* and *Tremillum* (*p* < 0.01) and significantly decreased the abundance of *Heterobacteria* (*p* < 0.01). This study can provide a theoretical basis for the replacement of traditional non-dairy creamer and the development of the walnut industry.

## Figures and Tables

**Figure 1 molecules-29-04469-f001:**
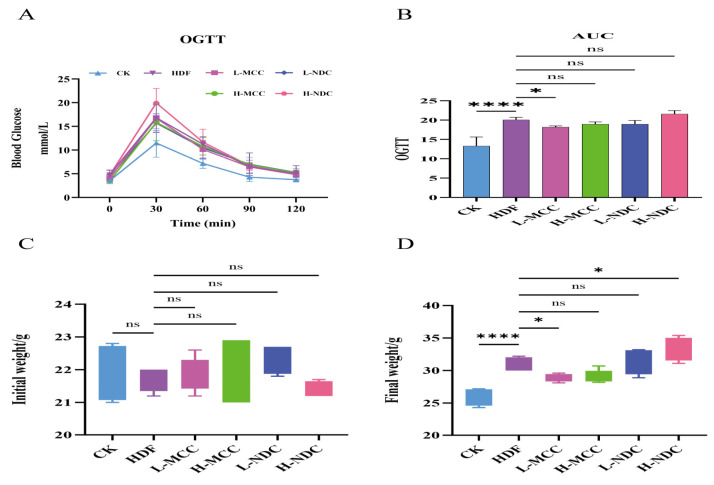
Effect of different non-dairy creamer on glucose tolerance and body weight in hyperlipidemic mice. (**A**) OGTT diagram of glucose tolerance. (**B**) Area under the glucose tolerance curve. (**C**) Initial weight of mice. (**D**) Final weight of mice (* *p* < 0.05, **** *p* < 0.0001, and ns for *p* > 0.05, not significantly different).

**Figure 2 molecules-29-04469-f002:**
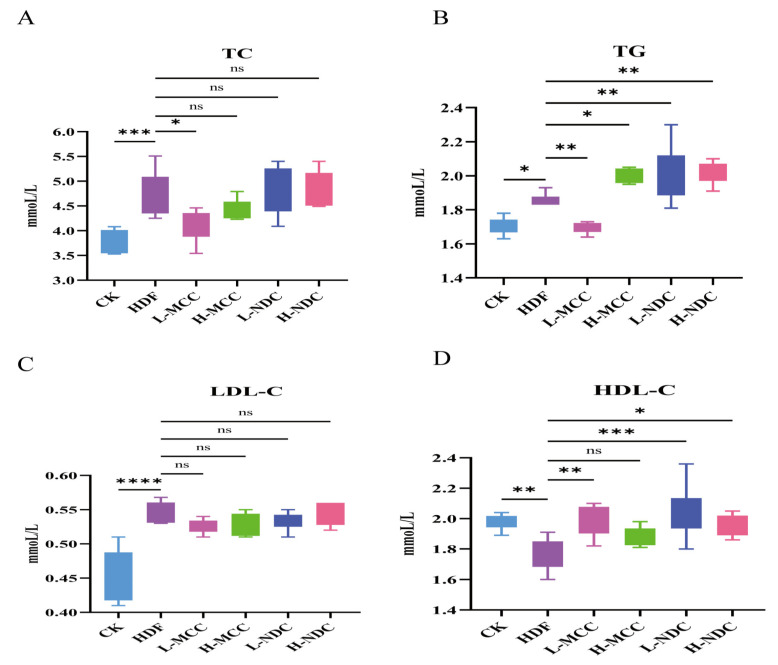
Effect of different non-dairy creamer on serum indices in high-fat mice. (**A**) TC; (**B**) TG; (**C**) LDL-C; (**D**) HDL-C (* *p* < 0.05, ** *p* < 0.01, *** *p* < 0.001, **** *p* < 0.0001, and ns for *p* > 0.05, not significantly different).

**Figure 3 molecules-29-04469-f003:**
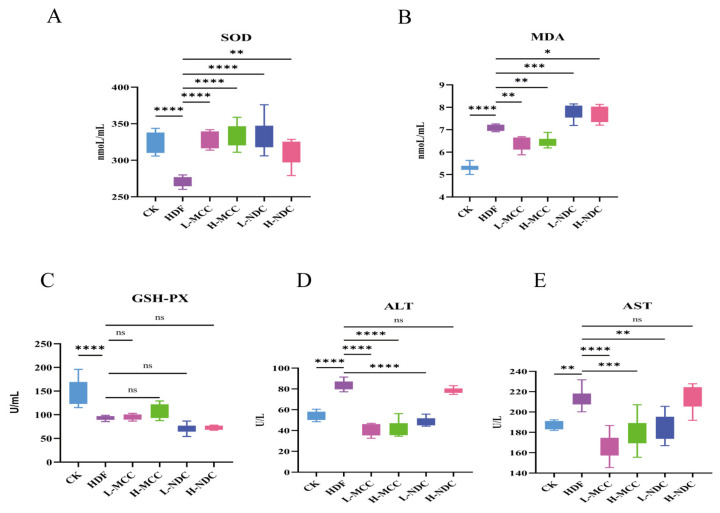
Effect of different non-dairy creamers on antioxidant indices and enzyme activities in high-fat mice. (**A**) SOD; (**B**) MDA; (**C**) GSH-PX; (**D**) ALT; (**E**) AST (* *p* < 0.05, ** *p* < 0.01, *** *p* < 0.001, **** *p* < 0.0001, and ns for *p* > 0.05, not significantly different).

**Figure 4 molecules-29-04469-f004:**
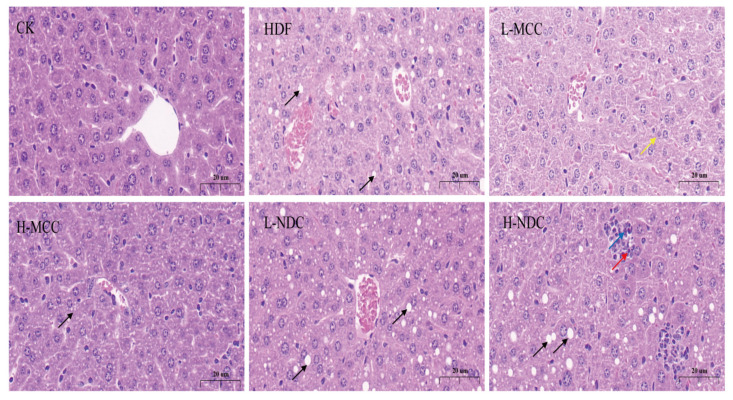
Effect of different non-dairy creamers on mouse hepatocytes. Magnification: 60.0×.

**Figure 5 molecules-29-04469-f005:**
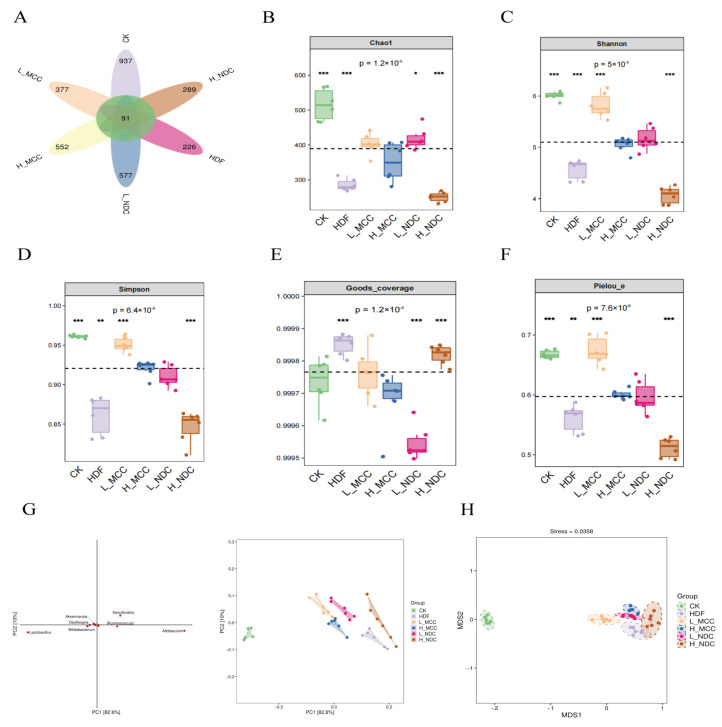
Effects of non-dairy creamer on intestinal flora diversity in mice. (**A**) Venn diagram, (**B**) Chao1 index in α-diversity analysis, (**C**) Shannon index, (**D**) Simpson index, (**E**) Goods_coverage index, (**F**) Pielou_e, (**G**) principal coordinate analysis (PCoA), (**H**) non-metric multidimensional scale of intestinal flora (NMDS). Values are expressed as mean ± SD (*n* = 6) (* *p* < 0.05, ** *p* < 0.01, *** *p* < 0.001, and ns for *p* > 0.05, not significantly different).

**Figure 6 molecules-29-04469-f006:**
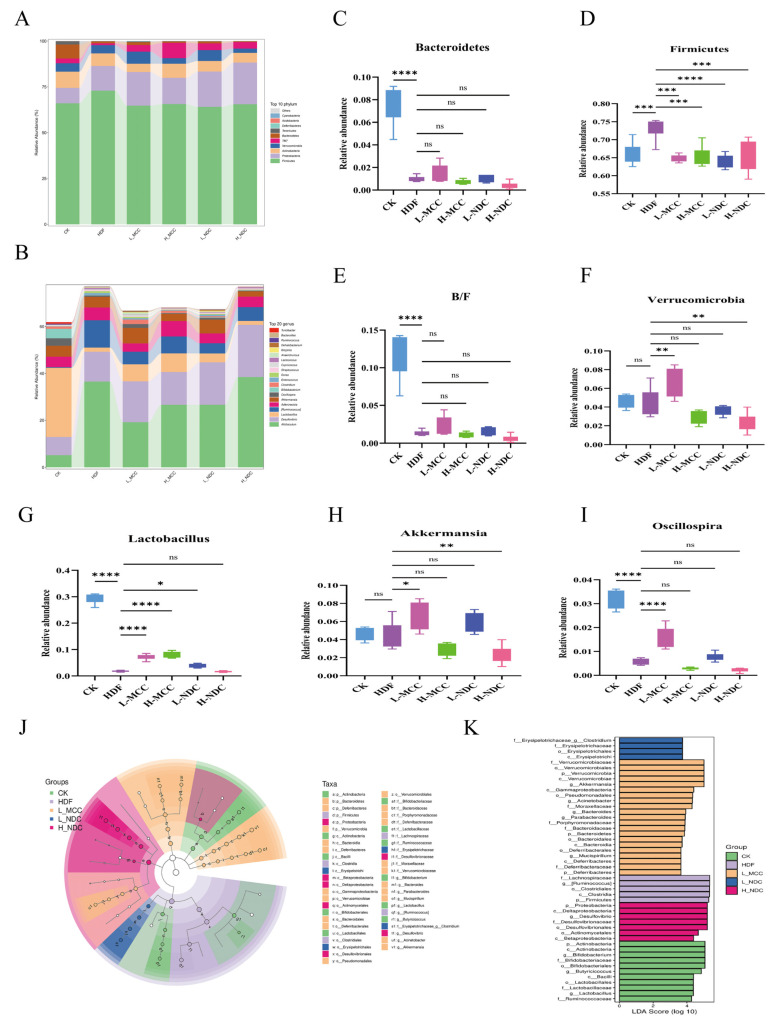
Effects of non-dairy creamer on intestinal flora in mice. (**A**) Phylum level, (**B**) genus level, (**C**) *Bacteroidetes*, (**D**) *Firmicutes*, (**E**) *B/F* ratio, (**F**) *Verrucomicrobia*, (**G**) *Lactobacillus*, (**H**) *Akkermansia*, (**I**) *Oscillospira*, (**J**) LEfse evolutionary branching diagram, (**K**) histogram of LDA scores. Values are expressed as mean ± SD (*n* = 6) (* *p* < 0.05, ** *p* < 0.01, *** *p* < 0.001, **** *p* < 0.0001, and ns for *p* > 0.05, not significantly different).

**Figure 7 molecules-29-04469-f007:**
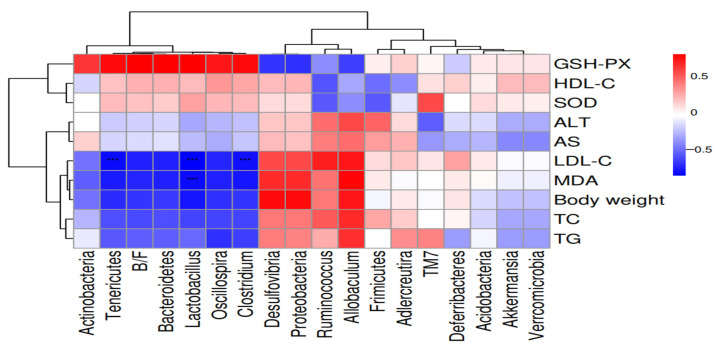
Correlation analysis heat map of intestinal microbiome and obesity-related parameters. Values are expressed as mean ± SD (*n* = 6) (*** *p* < 0.001).

**Table 1 molecules-29-04469-t001:** Effects of walnut non-dairy creamer on body weight and food intake of high-fat mice.

Group	Initial Weight (g)	Final Weight (g)	Food Intake (g/w)
CK	21.77 ± 0.33	26.70 ± 0.52 ^d^	140 ± 8.84 ^a^
HDF	21.63 ±0.33	31.15 ± 0.41 ^b^	103.98 ± 8.55 ^b^
L-MCC	21.90 ± 0.21	28.95 ± 0.24 ^c^	98.33 ± 7.16 ^b^
H-MCC	21.85 ± 0.36	28.63 ± 0.28 ^c^	100.19 ± 13.02 ^b^
L-NDC	21.68 ± 0.45	29.32 ± 0.38 ^c^	102.33 ± 6.18 ^b^
H-NDC	21.40 ± 0.21	33.55 ± 0.72 ^a^	100.95 ± 11.24 ^b^

Note: Values with different superscript letters in each column are statistically significant (*p* < 0.05).

**Table 2 molecules-29-04469-t002:** Feeding and gavage schemes of mice in different treatment groups.

Number	Groups	Quantities	Gavage Program
CK	CK	10	Normal saline
HDF	HDF	10	Normal saline
L-MCC	L-MCC	10	MCC (250 mg/kg)
H-MCC	H-MCC	10	MCC (1000 mg/kg)
L-NDC	L-NDC	10	NDC (250 mg/kg)
H-NDC	H-NDC	10	NDC (1000 mg/kg)

## Data Availability

The data presented in this study are available upon request from the corresponding author.

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
