# Peer review of "Regulation of Cecal Microbiota and Improvement of Blood Lipids Using Walnut Non-Dairy Creamer in High-Fat Mice: Replacing Traditional Non-Dairy Creamer"

_molecules, 2024, doi:10.3390/molecules29184469_

Round 1
Reviewer 1 Report
Comments and Suggestions for Authors
Comments are listed in the attached Word document

Comments on the Quality of English LanguageIt is included in the Word document. Language editing is needed.
Author Response
Dear Editor,
Thank you for giving us an opportunity to revise our manuscript, we appreciate editor and
reviewers very much for their positive and constructive comments and suggestions on cur
manuscript entitled “Walnut non-dairy creamer regulation of cecal microbiota and improvement of
blood lipids in high-fat mice: replacing traditional non-dairy creamer ” (Manuscript Number:
molecules-3135716). We have studied the comments carefully and have made revision in this
paper. We responded to reviewers' comments, Moreover, the revised paragraphs sentences are
labeled in red.
The revised paragraphs are labeled in red
Reviewer
(1) Line 398 “stored at - 80 °C until analyzed” – This seam as the end of the sentence, but the
punctuation mark is missing, if I am not mistaken?
Response 1: Thank you very much for your review comments! Line 373 - I have added
punctuation to the appropriate sections of the manuscript.
(2) Lines 415-418 – in the reviewed version of the manuscript, the formula and the text are not
well positioned, or I did not understand what authors aimed to show (image below).
Response 2: Thank you very much for your review comments! I have made changes to the
formulas and text.
(3) Line 422 – “A portion of mouse liver tissue was fixed in 4% paraformaldehyde fixative, …”
Remark: in the subsection 3.3. there is a sentence (line 400) saying thet liver tissue was fixed
in 10% formalin. Both dilutions would be correct from the point of methodology, but please
make the text uniform (Line 400: “in addition, a portion of the liver tissue was fixed in 10%
formalin and 400”).
Response 3: Thank you very much for your review comments! Line 395 - I have revised and
standardised the language used in the manuscript. The methods used in the manuscript are all 10%
formalin.
(4) Lines 423-424 – “ Hematoxylin-eosin(HE), and the sections were observed and photographed
using a light microscope.” The letter “H” in the word hematoxylin should be “h”, and from my
version of the manuscript, there is no space between word “ eosin” and the bracket “ (HE) ” .
Include exact type of microscope and camera used, as well as magnification under which the
tissue was photographed.
Response 4: Thank you very much for your review comments! Line 395 - I have changed the“H”
to“h”in the corresponding part of the manuscript and added spaces. In addition, I have added
the type and magnification of the microscope in “ 3.6 Histopathological Observation of Liver
Tissue”and Figure 3 of the manuscript.
(5) Line 426 – “Total DNA was collected from cecum samples using a DNA extraction kit to…” -
Perhaps it should say “ cecum content samples” in order not to get the wrong impression that
the tissue of the cecum was analyzed (as an organ) and not the intestinal contents. In section
3.3. this was clearly stated, I admit, but here it should be clarified too.
Response 5: Thank you very much for your review comments! Line 402 - I have made changes to
the “3.7” section of the manuscript.
(6) Table 1 – precisely indicate which superscript letter indicates significant difference between
what groups!!! This way presented and Table legend are unclear. Line 104 – “the AUC area
value of the HFD groupwas ” – typing error, please check the entire text, given the many
existing typos
Response 6: Thank you very much for your review comments! I have made changes to the
labeling of Table 1. Line 96- I have revised.
(7) Figure 1 needs improvement:
(1) Graphs B-D - there is a mark “ns” on the figure, but it is not interpreted in the legend. I
believe it means “non-significant”, but it would be necessary to address to it in the figure
legend.
(2) Figure legend (line 115) – “ …(C) original body weight of mice; (D) original body weight
of mice” – instead of original body weight it should say initial body weight, and Fig1-D is
not “original” or initial, but final body weight, as it is indicated on the figure itself.
(3) Figure legend (lines 115-116) - “ * denotes significance, and the more means the more
significant difference ” – this formulation is absolutely unacceptable for a manuscript
submitted to a Journal such as Molecules. If a certain values have different5level of statistical
significance, it must be clearly and differently labeled. Perhaps * -0,05, **- 0,01, *** - 0,001
or any other way.
(4) In this figure legend, but also throughout the manuscript it appears that there is a space missing
after the end of one sentence and the beginning of following, which makes it harder to read
and folow.
(5) name the Y axis on the graphs C and D, as it was done for A and B.
(6) since the X axis is named as Time (min), the axis marks should be numeric only (0, 30, 60,
90...) without the “min” following each value
Response 7: Thank you very much for your review comments! Line 107-108 -(1-3) In response to
these three questions, I have added and revised the labeling of Figure 1 in the manuscript. (4) I
have added spaces to the figure 1 labelling. (5) I have made changes to Figures 1C-D. (6) I have
made changes to Figure 1A.
(8) Lines 118-120 – sentence need rephrasing. Line 105 vs line 123 (just an example) – noting of
P values is somewhere with spacing and somewhere without. P < 0.01/ P<0.01.
Response 8: Thank you very much for your review comments! I have revised the sentences in
lines 111-114 of the manuscript. In addition, uniform changes have been made regarding the
spacing of p-values in the manuscript.
(9) Table 2 is in fact Figure 2? Individual graphs are to small and hard to interpret. This figure
should be displayed as two figures, perhaps one representing lipid and liver function
parameters and another with MDA and antioxidative enzymes. Same as for Figure 1 -”ns”
lbel is not explained; different level of statistical significance must be presented with
different type of labels on the graph and explanation “*:denotes significance, and the more
means the more significant difference.” in the Legend is unacceptable.
Response 9: Thank you very much for your review comments! Yes, that one should be figure 2
and I have made changes. I have split figure 2 into two figures. Also, regarding the error with the
“ns”and *, I've made a change in question 7.
(10) Line 146 – “SOD level of all kinds of mice” – all mice are of the same kind, but perhaps
authors wanted to say of all groups? I believe that this kind of mistakes are result of low
quality translation from native language, and therefore the text need editing by a proficient
language expert
Response 10: Thank you very much for your review comments! Yes, I strongly agree with your
comments. Therefore, I have made the changes as you have suggested in line 139.
(11) Line 147-149 –“There was no significant difference in the level of SOD between the MCC
groups (P>0.05), and the level of SOD between the NDC groups was significantly decreased
with the dosage of the elevation, SOD levels decreased significantly (P<0.01).” Sentence is
unclear and hard to understand. One more example of need for expert language editing.
Response 11: Thank you very much for your review comments! I apologise for any distress
caused to you through my negligence. In response to the issue, I have had the sentence reworked
line 140-142 by a professional teacher I consulted.
(12) Line 151-152 – “MDA is a membrane lipid peroxidation product that indirectly reflects the
damage of hepatocytes under lipid peroxidation and oxidative stress in fatty liver patients in
vivo [26]”– MDA is a marker of lipid peoxidation damage to membranes of any cell, not
just hepatocytes, and from any reason not just NAFLD in vivo. Rephrase the sentence.
Response 12: Thank you very much for your review comments! Line 143-145 - I have edited and
revised the text.
(13) Line 153 – “As can be seen from Figure 2F, the HDF group significantly increased the MDA
content of high-fat mice compared with the CK group (P<0.01).” – It is not the group that
have raised the MDA levels to high fat mice, it is the high fat diet that have increased MDA
in mice of HDF group. Please be more precise and careful when writing.
Response 13: Thank you very much for your review comments! I apologize for the trouble I
caused you due to a writing error, I have rewritten the paragraph in line 145-147.
(14) Line 156 - .. between the two kinds (P>0.05).´- again, use the term groups not kinds. I will
not note further use of this word instead of group or some more appropriate term. Please
review the text and correct it at all sentences
Response 14: Thank you very much for your review comments! Line 151 - I have made changes
in the manuscript. In addition, I have checked the full text and corrected it accordingly.
(15) Line 156-159 – ” Zhang [25] found that hazelnut oil microcapsules also had the effect of
reducing the level of lipid peroxidation products, which is consistent with our findings. And
all NDC groups significantly increased the MDA content. In conclusion, MCC can enhance
the antioxidant activity in high-fat mice. ” – The underlined sentence, can not start with
“and”, it seems like it is a remnant of some other sentence, unfinished. Whole paragraph to
which these sentences are taken, refers to MDA as a lipid peroxidation damage marker, so the
conclusion can not be about antioxidant activity of the mice. You may conclude about
prooxidative impact of high fat diet, or impact of walnut oil in MDA 7and
alleviating/preventing lipid peroxidation. The antioxidative capacity can be commented based
on level of ROS or antioxidative enzymes/substances.
Response 15: Thank you very much for your review comments! I couldn't agree more with your
comments. Therefore, I have revised this paragraph in line 149-152.
(16) Line 165 – “ With the HDF group, the GSH-PX content was 165 reduced in both NDC
groups.” - rephrase the sentence it is unclear, language editing.
Line 179 –“The results showed that MCC could effectively reduce liver injury in highfat
mice. effect on ALT and AST activities (P>0.05). The results showed that MCC 180 could
effectively reduce liver injury in high-fat mice.” The underlined sentence is just pending as
if it is a leftover of some other sentence? The sentence in Italic is a duplicate of the first
sentence?
Response 16: Thank you very much for your review comments! Line 155-169 - I have revised
this paragraph . Line 178-179 - I am very sorry for making such a simple mistake due to my
negligence, and I have corrected this sentence.
(17) Line 182 – “2.6. Effect of different non-dairy creamer on H&E staining of liver cells in highfat
mice” – creamer can not affect the HE staining.
Line 186 – ” In the HDF group, the hepatocytes were loosely arranged and disorganised
without clear boundaries, and the liver hepatocytes were covered with fat particles of varying
sizes (black arrows) in diffuse steatosis.” - Not having clear boundaries would indicate a
necrosis of hepatocytes. Please consult histologist or pathologist on this finding. Also,
hepatocytes can not be covered by fat particles. Lipid droplets, vesicles can be found inside
the hepatocytes as a sign of steatosis. This section needs to be reviewed by histologist or
pathologist and as such it must be included as one of the authors.
Response 17: Thank you very much for your review comments! Line 177 – I have amended the
title as follows:“2.6. Effect of different non-dairy creamer on liver cells in high-fat mice”. Line
186 – I couldn't agree more with your comments. I would like to indicate this part of the content is
tested and analyzed by a professional organization (organization name: Yunnan Huiteng
Biotechnology Co., LTD.). In addition, this part of the content is guided and analyzed by the
corresponding author of the manuscript and experts.
(18) Line 190 – “ led to the development of steatosis and tissue lesions. ” - no other tissue
lesion except mild steatosis is not visible in those images.
Line 190 – ” The above results indicate that MCC can improve the fibrosis phenomenon in
the livers of high-fat mice, and it has a certain positive intervention effect on the degeneration
of fat in the livers of mice, with a certain hypolipidemic function, and it is more effective
than NDC.” - by no means the presented results can indicate the improvement of fibrosis.
Authors must firstly demonstrate that the fibrosis occurred in HDF group. To do so, you must
use some specialized histochemical staining. This section needs to be reviewed by histologist
or pathologist and as such it must be included as one of the authors. Histology is of great
value in experimental research, but it takes great caution and expertise in presenting and
interpreting it.
Response 18: Thank you very much for your review comments! I couldn't agree more with your
comments. I have removed the tissue lesion from the manuscript. Line 185 – 194 - I have
consulted relevant experts and re-analyzed and corrected this sentence.
(19) Line 205 – “Table 3.Effects of different non-dairy creamer on pathological sections of liver
of high-fat mice” - This is not a Table, it is a microphotograph and I believe that according
to Instructions to authors it should be labeled as Figure.
Response 19: Thank you very much for your review comments! Yes, this is a figure, not a table. I
have made changes.
(20) Figure 3 (mislabeled image of liver tissue) - Editing tissue microphotographs also must be
precise – there are no indications of yellow, blue and red arrows? individual images are blur,
so please replace them with higher resolution images, and preferably of higher magnification.
In the title of the image, you must state the staining method and the magnification of the
tissue. Effects of different non-dairy creamer on pathological sections of liver of high-fat
mice”- creamer does not affect tissue sections, it affects the liver tissue, and here authors
present images of liver tissue of all groups so it can not say“of high fat mice”.
Response 20: Thank you very much for your review comments! I am very sorry for the trouble I
caused you, due to my negligence caused some content omission, I have added the appropriate
content in the manuscript and detailed the arrows used. I have indicated the magnification in the
caption of the image. In addition, the method used I have added in the heading
“3.6.Histopathological Observation of Liver Tissue.”
(21) Line 214 – Authors refers to Fig 4A, but there is no Figure title or number, although the
image is placed in the manuscript. Individual images / graphs in the supposed Figure 4 are
very small and hard to read/interpret (particularly 4G and 4H).
Response 21: Thank you very much for your review comments! I am very sorry for the trouble I
caused you, due to my negligence caused some content omission, I have added the appropriate
content in line 238-242.
(22) Line 221 –“3.7.2 Alpha Diversity Analysis – this should be subsection 2.7.2.
Line 237 –“suggesting that the high-fat diet significantly altered the composition of gut
microorganisms in normal mice” - which mice are not normal? Do authors think of control
group?
Response 22: Thank you very much for your review comments! Line 212 - I have modified this
part of the content. Line 228 - 230 - Yes, the author's intent was to compare to a control group. I
apologize for troubling you due to unclear expression. I have rewritten this passage.
(23) Lines 247-270 – there lines are blank, I do not know for what reason.
Line 271 – “2.7.4 phylum level analysis” – the title should have initial letter capitalized
Figure 5 (Figure on the page 10 I guess), is present in the manuscript but has no Title or
legend, and individual graphs are very small and blur.
Response 23: Thank you very much for your review comments! I'm really sorry for making such
a stupid mistake. I have removed the spaces in the manuscript. Line 246 - I have capitalized the
first letter of the title and added the legend accordingly. In addition, I made a clarity adjustment to
the picture of the manuscript.

Reviewer 2 Report
Comments and Suggestions for Authors
Dear Authors:
Thank you for your manuscript about "Walnut oil microcapsules regulation of cecal microbiota and improvement of blood lipids in high-fat mice: replacing traditional non-dairy creamer".
The design and experiment with both analysis and discussion are well presented, however for regular reader, it is difficult to follow the long sentences you produced.
For example; in The first paragraph first sentence is more than 40 words, followed by another long sentence.
In the results;
As can be seen from Fig- 238ure 4H, all MCC groups could be separated from the HDF group and were farther away, 239
while the NDC group was closer to the HDF group and the H-NDC group partially over- 240
lapped with the HDF group, suggesting that the intestinal microbial composition of the 241
H-NDC group was closest to that of the HDF group, whereas the MCC group was closest 242
to the CK group and the L-MCC was closest to the CK group, suggesting that the MCC 243
intervention was able to effectively improve the hyperlipidemic mice's intestinal flora 244
composition.
This can't be one sentence.
so please check all long sentences with more than one idea and keep it one idea for one sentence.
Regards
Comments on the Quality of English LanguageNeeds some revesion all through the manuscript.
Author Response
Dear Editor,
Thank you for giving us an opportunity to revise our manuscript, we appreciate editor and reviewers very much for their positive and constructive comments and suggestions on cur manuscript entitled “Walnut oil microcapsules regulation of cecal microbiota and improvement of blood lipids in high-fat mice: replacing traditional non-dairy creamer” (Manuscript Number: molecules-3135716). We have studied the comments carefully and have made revision in this paper.We responded to reviewers' comments, Moreover, the revised paragraphs sentences are labeled in blue.
The revised paragraphs are labeled in blue.
Reviewer
- In The first paragraph first sentence is more than 40 words, followed by another long sentence.
Response 1: Thank you very much for your review comments! I have revised this paragraph. In addition, I also modified other long sentences in the manuscript and marked them with blue.
- Lines-240-244 - while the NDC group was closer to the HDF group and the H-NDC group partially overlapped with the HDF group, suggesting that the intestinal microbial composition of the H-NDC group was closest to that of the HDF group, whereas the MCC group was closest to the CK group and the L-MCC was closest to the CK group, suggesting that the MCC intervention was able to effectively improve the hyperlipidemic mice's intestinal floracomposition. This can't be one sentence.so please check all long sentences with more than one idea and keep it one idea for one sentence.
Response 2: Thank you very much for your review comments! 230-237 - I am very sorry for the trouble caused to you due to the writing error. I have rewritten this passage in the manuscript.
Note: I also uploaded the attachment.

Round 2
Reviewer 1 Report
Comments and Suggestions for Authors
Detailed comments are given in the word document attached below.

Comments on the Quality of English LanguageThe whole manuscript (not just the parts which I specifically indicated in the first review) should be inspected by a language expert
